# m^6^A RNA Methylation in Marine Plants: First Insights and Relevance for Biological Rhythms

**DOI:** 10.3390/ijms21207508

**Published:** 2020-10-12

**Authors:** Miriam Ruocco, Luca Ambrosino, Marlene Jahnke, Maria Luisa Chiusano, Isabel Barrote, Gabriele Procaccini, João Silva, Emanuela Dattolo

**Affiliations:** 1Stazione Zoologica Anton Dohrn, Villa Comunale, 80121 Naples, Italy; luca.ambrosino@szn.it (L.A.); marialuisa.chiusano@szn.it (M.L.C.); gpro@szn.it (G.P.); dattolo@szn.it (E.D.); 2Department of Marine Sciences, Tjärnö Marine Laboratory, University of Gothenburg, SE-45296 Strömstad, Sweden; marlene.jahnke@gu.se; 3Department of Agricultural Sciences, University of Naples Federico II, 80055 Portici (NA), Italy; 4CCMar—Centre of Marine Sciences, University of Algarve, Campus of Gambelas, 8005-139 Faro, Portugal; ibarrote@ualg.pt (I.B.); jmsilva@ualg.pt (J.S.)

**Keywords:** marine plants, epitranscriptome, m^6^A-methylation, circadian clock, RNA methyltransferases, RNA demethylases, daily cycle, photoperiod, latitude

## Abstract

Circadian regulations are essential for enabling organisms to synchronize physiology with environmental light-dark cycles. Post-transcriptional RNA modifications still represent an understudied level of gene expression regulation in plants, although they could play crucial roles in environmental adaptation. N^6^-methyl-adenosine (m^6^A) is the most prevalent mRNA modification, established by “writer” and “eraser” proteins. It influences the clockwork in several taxa, but only few studies have been conducted in plants and none in marine plants. Here, we provided a first inventory of m^6^A-related genes in seagrasses and investigated daily changes in the global RNA methylation and transcript levels of writers and erasers in *Cymodocea nodosa* and *Zostera marina*. Both species showed methylation peaks during the dark period under the same photoperiod, despite exhibiting asynchronous changes in the m^6^A profile and related gene expression during a 24-h cycle. At contrasting latitudes, *Z. marina* populations displayed overlapping daily patterns of the m^6^A level and related gene expression. The observed rhythms are characteristic for each species and similar in populations of the same species with different photoperiods, suggesting the existence of an endogenous circadian control. Globally, our results indicate that m^6^A RNA methylation could widely contribute to circadian regulation in seagrasses, potentially affecting the photo-biological behaviour of these plants.

## 1. Introduction

Epigenetic modifications include all changes that provide heritable phenotypes while not altering the DNA sequence itself, such as chemical modifications of DNA (e.g., methylation) and its associated proteins (e.g., histone modifications), or changes caused by small RNA molecules (e.g., gene silencing by noncoding RNAs) [1]. These multiple epigenetic mechanisms are critical for regulating the chromatin condensation state and directly affect gene expression during organismal development and in response to environmental stressors [2,3,4].

In parallel, the term ‘epitranscriptomics’ comprises all modifications of RNA molecules that do not affect their sequence [5]. Similar to epigenetic changes involving the genome structure, RNA modifications are emerging as fundamental players in the field of post-transcriptional regulation of gene expression in both plants and animals, and are attracting a comparable interest to that of DNA and histone modifications [5,6,7,8]. Furthermore, emerging studies are highlighting the roles of RNA methylation as an epigenetic marker for RNA-dependent inheritance [9,10] and the induction of a stable phenotypic variation [11].

More than 150 different and naturally occurring chemical modifications of different degrees and topologies have been detected on mRNAs, tRNAs and rRNAs [12,13]. On mRNAs, which generally tend to be less densely modified than other RNAs, few chemical modifications have been recognized, with *N*^6^-methyladenosine (m^6^A) being the most frequent [14]. The presence of such modifications and their relative abundance can affect mRNA expression at multiple stages, including splicing, nucleus-to-cytoplasm export, turnover, and translation ability, via structural rearrangements and shaping RNA-protein interactions [15,16,17,18,19]. Unlike in mammals, in plants m^6^A sites are enriched not only around the stop codon and within 3′ untranslated regions (3′ UTRs), but also around the start codon of CDS [20,21], indicating that m^6^A modifications can regulate all stages of mRNA life.

Proteins responsible for adding, removing, and interpreting RNA methylation marks are designated as “writers” (methyltransferases, MTs), “erasers” (demethylases, DMTs), and “readers” (RNA-binding proteins), respectively. They have been identified and well-characterized in animals [19,22,23,24,25], whereas only recently functional orthologs have been discovered in terrestrial model plants, such as *Arabidopsis* and rice [26]. Although many studies now point to the importance of RNA methylation and the essential roles writers, readers, and erasers might play in plant development and stress response [21,27,28,29,30,31], the functionality of most of these proteins remains unclear.

In *Arabidopsis*, at least five proteins have been identified as members of the multicomponent m^6^A writer complex, namely MTA (ortholog of METTL3 in animals), MTB (ortholog of METTL14), FIP37 (ortholog of WTAP), VIRILIZER (ortholog of WIRMA), and HAKAI [26]. In animals, the primary enzyme with an RNA methyltransferase activity is METTL3, while METTL14 has a supporting role forming a heterodimer with METTL3 [26]. The other two writer components, WTAP and VIRMA, play a role as a heterodimer stabilizer [24] and in guiding the methyltransferase complex to target regions on mRNAs, respectively [32]. The role of HAKAI, a recently identified E3 ubiquitin ligase that is present as an additional m^6^A writer component in *Arabidopsis,* is still to be uncovered [33]. The removal of m^6^A modifications is carried out by AlkB RNA demethylases (ALKBH) that catalyse the m^6^A demethylation in a α-ketoglutarate (α-KG) and Fe^2+^-dependent manner [34]. The *Arabidopsis* genome contains several putative m^6^A erasers belonging to the ALKBH family, among which only two (i.e., ALKBH9B and ALKBH10B) have been functionally characterised during viral infection and flowering [26]. RNA-binding proteins (RBPs), which recognise methylated mRNAs and determine their expression, carry out the interpretation of RNA methylation marks [35].

Circadian rhythms are essential for enabling plants to synchronise endogenous physiological processes such as photosynthesis and metabolism with environmental changes occurring on daily and seasonal scales [36,37], thus increasing individual fitness and survival [38,39]. Such biological rhythms of approximately 24 h are generated and maintained by circadian clocks. A complex network of interlocked feedback loops composes the core of the plant clock (or oscillator), where clock genes act both within the oscillator and in clock input and output signalling pathways [37]. The plant circadian clock is composed of at least three intertwined transcriptional loops, which represent a fundamental part of the clock regulatory system. Likewise, multiple post-transcriptional regulative mechanisms play a critical role in the regulation of the clock network [37].

Recent studies have shown that epigenetic and epitranscriptomic regulations affect the expression of clock components in several organisms [28,40,41], but only few studies have been conducted on the epigenetic and post-transcriptional controls of circadian clock functions in plants [42,43,44,45] and almost none on the role of RNA methylation (m^6^A) in regulating clock-related genes [46,47]. In mammals, it has been demonstrated that a large number of transcripts encoding core clock components, as well as clock output genes, are enriched in m^6^A methylation sites [40,48] and that changes in their m^6^A levels can affect circadian rhythms, cellular growth, and ultimately survival [49]. Furthermore, the expression of the RNA methyltransferase METTL3 was inversely correlated with the circadian rhythm duration, and its knockdown resulted in a delayed processing of specific clock-related transcripts [48]. Finally, it has been demonstrated that m^6^A RNA modifications per se are under the control of circadian regulators [50]. Interestingly, it has been observed in *A. thaliana* that a large fraction of m^6^A-containing mRNAs was associated with specific pathways involving the chloroplast [21], suggesting an active role of m^6^A in the regulation of circadian-regulated processes, such as photosynthesis, in plants.

Seagrasses are a unique group of flowering plants with a key ecological importance, which have adapted to a marine lifestyle [51,52]. The successful colonisation of marine environments was contingent upon a number of critical physiological and biochemical adaptations, based on specific genomic losses and gains [51,53,54], and adaptive changes in sets of genes associated with central biological pathways, such as photosynthesis [55]. The exposure to extreme underwater light and temperature regimes influenced seagrasses’ circadian clock components and photoreceptors, in terms of the presence/absence of related genes [51] and their regulation at daily and seasonal scales [56]. A first evidence of differential daily expression patterns of photosynthesis and circadian clock-related genes in seagrass populations has come from *Posidonia oceanica* across a bathymetric gradient [56,57]. However, how epigenetic and epitranscriptomic mechanisms (e.g., m^6^A RNA methylation) could affect seagrasses’ circadian clock and regulate the expression of its components across various temporal and spatial scales, and the differences with respect to such mechanisms in other angiosperms, is completely unknown so far.

To begin filling this research gap, we here (i) provide a first inventory of genes involved in m^6^A RNA methylation in seagrasses, exploiting publicly available genome and transcriptome data from *Zostera marina* and *Cymodocea nodosa*, (ii) investigate daily changes in RNA methylation in the two seagrass species through global *N*^6^-methyl-adenosine (m^6^A) quantification, and (iii) quantify the transcript levels of members of the core m^6^A writer and eraser complexes via RT-qPCR over a 24-h cycle. Both intra- and interspecific comparisons were established for global m^6^A profiles and expression changes of RNA-methylation-related genes. Specifically, the differential behaviour of *C. nodosa* vs. *Z. marina* (interspecific) was explored at the same location (i.e., Ria Formosa lagoon, Faro—Portugal), while an interpopulation comparison (intraspecific) was explored comparing the response of *Z. marina* growing at different latitudes, i.e., Faro—Portugal (Lat. 37.019356) vs. Tjärnö—Sweden (Lat. 58.879724), characterized by large differences in the photoperiodic regimes.

## 2. Results and Discussion

### 2.1. First Identification of RNA Methyltransferase (Writers) and Demethylase (Erasers) Genes in Seagrasses

Genes encoding for m^6^A writer and eraser proteins have been identified in terrestrial plants, after being characterized in animals [24]. Here, we provided a first inventory of these genes in marine plants in terms of the presence/absence and expression patterns across species and under different conditions. We screened the protein complement of the *Z. marina* genome, the first seagrass species for which a full genome sequence has been obtained [51], and the one of a previously published transcriptome of *C. nodosa* [58], together with the protein complement of *Arabidopsis thaliana* and *Oryza sativa* genomes, to identify writer and eraser proteins through a comparative approach. The comparative analysis of all the considered species led to the construction of networks of ortholog and paralog proteins. In parallel, to identify proteins related to writer and eraser functions in the networks, we performed a sequence similarity search of the protein collections of the four species versus the Swiss–Prot database. In general, all m^6^A writers identified in terrestrial plants were also present in the analysed seagrass species (Table 1). In the *Z. marina* genome, single proteins represented MTA, MTB, and HAKAI, while FIP37 resulted in a gene family of six members (Table 1, Figure 1 and Figure 2). While *Arabidopsis* has two isoforms of VIRILIZER, *Z. marina* only possesses one VIRILIZER protein. In the *C. nodosa* transcriptome, two transcripts for MTA and four for MTB were identified, together with seven transcripts for FIP37, three for VIRILIZER, and one encoding for HAKAI (Table 1, Figure 1 and Figure 2).

Alfa-ketoglutarate-dependent dioxygenase (AlkB) homolog (ALKBH) proteins act as erasers, carrying out the removal of methylation marks on mRNAs. In *Z. marina*, only one gene encoding for ALKBH9B was present, in contrast with the presence of ten isoforms encoded by three genes with the same function in *Arabidopsis* [26]. Both species have three genes encoding for ALKBH10B, with one gene encoding for two isoforms in *A. thaliana* (Table 1 and Figure 3). In *C. nodosa*, there were at least 13 transcript variants encoding for ALKBH9B and 20 for ALKBH10B (Table 1 and Figure 3). By observing the topology of this network, a clear split of ALKBH9B and ALKBH10B members can be noticed (Figure 3). The clustering of these two subfamilies within the same network occurs because of the presence of the three *O. sativa* ALKBH9B and ALKBH10B proteins, probably common ancestors of the AlkB family that acted as a bridge between the two subfamilies. Although several putative eraser proteins have been identified in terrestrial plant species, only ALKBH9B and ALKBH10B have been functionally characterised [59,60], and our analysis has therefore only focussed on these two components.

Dedicated analyses (by multiple alignments, data not shown) confirmed that similar copies from *C. nodosa* sequences included in the considered networks have to be considered as products of alternative transcripts that are worth being further investigated as a result of the already published transcriptome assembly [58].

The protein sequences of the considered networks were used as templates for dedicated phylogenetic analyses. The phylogenetic trees, showing clear separations between the clustered leaves, confirm the topology of the constructed networks (Figure 1, Figure 2 and Figure 3).

The analysis of the gene family composition of the m^6^A complex identified a high number of RNA-methyltransferases, especially belonging to the FIP37 family, in marine plants with respect to the terrestrial counterparts. This result may suggest that m^6^A modifications could diversify because of their involvement in the responses to various biotic and abiotic stressors [31] in the marine environment. Besides this, as a recent study has highlighted the role of some m^6^A writers components, such as FIP37, in the embryonic plant development [61], we could speculate that the adaptation of seagrasses to the marine life style has required modifications, which are under epitranscriptomic control.

### 2.2. Interspecific Variations in the Daily Transcript Levels of Writers and Erasers and Global N^6^-Methyl-Adenosine (m^6^A) in Seagrasses

To explore the relevance of m^6^A RNA methylation in the regulation of seagrass daily rhythms, we analysed the transcript levels of core components of writer (MTA and MTB) and eraser (ALKBH9B) (Table 2) complexes along a 24-h cycle in both seagrass species. An interspecific comparison was done between *Z. marina* and *C. nodosa* growing in a mixed meadow at Ria Formosa lagoon, Faro—Portugal (Lat. 37). A multivariate analysis (two-way PERMANOVA) of gene-expression data (all three GOIs) for six selected time points (i.e., sunrise, solar noon, sunset, dusk, midnight, and dawn) highlighted a highly significant difference in the behaviour of the two species (*p*_(perm)_ < 0.001) and a significant species × time interaction (*p*_(perm)_ < 0.01) (Table 3). As depicted in Figure 4, in both species each GOI exhibited a dynamic daily regulation, although with asynchronous up- and downregulation peaks.

In *C. nodosa*, the dimer-forming genes MTA and MTB followed a clear overlapping pattern, with peaks of mRNA abundance during night hours and a maximum at midnight (0:00). Afterwards, the level of these transcripts started to decrease, reaching the negative peaks during light hours, i.e., between noon (13:00) and sunset (20:30) (Figure 4). ALKBH9B exhibited a very similar daily expression profile of RNA methyltransferases, with the peak of abundance at midnight and the lowest transcript level at solar noon (Figure 4). Overall, in *C. nodosa*, the daily expression pattern of MTA, MTB, and ALKBH9B was significantly positively correlated (MTA vs. MTB: *R* = 0.62, *p* = 0.005, *N* = 18; MTA vs. ALKBH9B: *R* = 0.94, *p* = 0.000, *N* = 18; ALKBH9B vs. MTB: *R* = 0.58, *p* = 0.011, *N* = 18). According to the pairwise results of PERMANOVA (Table 3), significant differences in the global expression of all three GOIs were found in C. nodosa for the contrasts “sunrise vs. solar noon” (*p*_(MC)_ < 0.05) and “sunrise vs. sunset” (*p*_(MC)_ < 0.05). The comparisons “midnight vs. solar noon” and “midnight vs. sunset” were almost significant (*p*_(MC)_ = 0.06) (Table 3).

In *Z. marina*, MTA and MTB followed a daily trend that was shifted with respect to *C. nodosa*, with peaks of mRNA abundance between solar noon and sunset, then decreasing during the night period, until reaching a negative peak at dawn/sunrise (Figure 4). ALKBH9B showed a more constant expression profile along the day, but still with a lower transcript level at midnight (Figure 4). Overall, in *Z. marina*, the daily expression pattern of MTA and MTB was significantly positively correlated (*R* = 0.86, *p* = 0.000, *N* = 18), as was that of MTB and ALKBH9B (*R* = 0.55, *p* = 0.023, *N* = 18). The pairwise PERMANOVA results for *Z. marina* highlighted significant differences in the global expression of all three GOIs in the contrasts “sunrise vs. sunset” (*p*_(MC)_ < 0.01) and “sunrise vs. dusk” (*p*_(MC)_ < 0.05). Almost significant results were found for “sunrise vs. solar noon” (*p*_(MC)_ = 0.08) and “dawn vs. sunset” (*p*_(MC)_ = 0.06) (Table 3).

A univariate analysis of individual targeted genes (two-way ANOVA) confirmed a highly different daily expression pattern between the two co-occurring species for both writer and eraser genes (*p* < 0.001 for MTA, MTB, and ALKBH9B), and a significant species × time interaction (*p* < 0.01 for MTA and ALKBH9B; *p* < 0.05 for MTB) (Table 4). The transcript levels of MTA and MTB were different between *C. nodosa* and *Z. marina* at all selected time points (see SNK results in Figure 4), whereas for ALKBH9B significant differences were only observed at midnight (see SNK results in Figure 4).

The gene-expression data were confirmed by the interspecific analysis of the daily pattern of the global N^6^-methyl-adenosine (m^6^A) level. A significant difference was found between seagrass species at the same geographic location (Table 5). Although *Z. marina* and *C. nodosa* exhibited a similar m^6^A level pattern throughout the day, this was shifted in time (Figure 4). In both species, the global m^6^A level started to decrease during light hours, from sunrise to sunset, then peaking at dusk in *C. nodosa* and at midnight in *Z. marina* (Figure 4).

Our results point to a possible circadian regulation of RNA methyltransferase and demethylase genes in both seagrass species, and consequently of their global m^6^A RNA levels, as observed in other organisms [46]. Notably, such daily rhythms were different for each species at the same geographic location, suggesting that they were not only regulated by environmental cues (i.e., light) but by endogenous species-specific signals. It is already known that marine angiosperms show circadian rhythms, as much of their biochemistry and physiology is temporally organised with respect to the oscillation of day and night [54,60,61]. The photosynthetic process is one of the main metabolic routes, for which the correct matching between endogenous clocks and day-night cycles is fundamental [62]. According to the daily variation in irradiance levels, seagrasses maintain a permanent and dynamic trade-off between photosynthetic efficiency and photoprotection [61,63]. Additionally, the respiratory activity can oscillate significantly during the day [54,60]. In the seagrass *Posidonia oceanica*, such physiological rhythms have been correlated with the expression patterns of related genes (e.g., photosynthesis and respiratory genes), which seem to be responsible for driving such changes throughout the day [54]. However, their upstream regulatory mechanisms, which could involve epigenetic and/or epitranscriptomic mechanisms [39,40,41,42], remain unknown.

Although the coupling of these processes deserves many further investigations, the m^6^A RNA rhythmicity observed here could play a role in regulating the expression patterns of genes involved in photosynthesis and other processes (such as respiration and sugar metabolism) or their upstream regulators (e.g., circadian-clock components) along the day/night cycle, thus affecting the photobiology of both *Z. marina* and *C. nodosa*. Photo-physiological parameters collected during the course of the experiment in Faro indicated that the two seagrass species displayed a different photo-physiological behaviour throughout the day, where *C. nodosa* exhibited a higher capacity to dissipate excess excitation energy than *Z. marina* [64], as well as a different soluble sugar content at predawn (04:00) and solar noon [63]. This confirms previously observed differences between *C. nodosa* and a close relative of *Z. marina* (i.e., *Zostera noltii*) [61]. Genes with m^6^A enrichment have been demonstrated to be highly enriched in chloroplast and photosynthetic constituents in *A. thaliana*, as is for instance the case with STN8, which is a protein kinase specific to the phosphorylation of D1, D2, and CP43 proteins of photosystem II [18]. This suggests a close relationship between m^6^A mRNA methylation and photosynthesis [18], which should also be explored in seagrasses, as it could be somehow responsible for the observed differences in the photo-physiology of the species.

### 2.3. Intraspecific Variations in the Daily Transcript Levels of Writers and Erasers and Global N^6^-methyl-Adenosine (m^6^A) in Seagrasses across Latitudes

Differences in the transcript levels of RNA methylation-related genes and m^6^A levels were also explored in the same species (*Z. marina*) growing at contrasting latitudes, with a very different photoperiod. In particular, an intraspecific comparison was done between *Z. marina* collected at Ria Formosa lagoon, Faro—Portugal (southern lat. 37.019356) and Tjärnö, Strömstad—Sweden (northern lat. 58.879724).

A multivariate analysis (two-way PERMANOVA) of gene-expression data (all three GOIs) for four selected time points (i.e., sunrise, solar noon, sunset, and midnight) highlighted significant effects of the factors time (*p*_(perm)_ < 0.05) and latitude (*p*_(perm)_ < 0.05) (Table 3). Considering the contribution of the multivariate dataset, larger differences in the abundance of GOIs were found for the contrasts “sunrise vs. sunset” (*p*_(perm)_ < 0.05) and “sunrise vs. solar noon” (*p*_(perm)_ = 0.08) at Faro and between midnight and sunset (*p*_(perm)_ = 0.08) at Tjärnö (Table 3).

MTA and MTB followed an overlapping daily profile in *Z. marina* across latitudes (correlation analysis; *R* = 0.87, *p* = 0.000, *N* = 12), whereas only for ALKBH9B was there a significant “latitude effect” (Table 4). According to a univariate two-way ANOVA, the level of the dimer-forming MTA/MTB was only affected by the time of the day (Table 4). At both latitudes, their abundance increased during light hours, from sunrise to sunset, where they reached the peak of expression. Afterwards, their level started to decrease again until midnight (Figure 5). At Faro, major differences were observed between sunrise and sunset (MTA: *p* = 0.05; MTB: *p* = 0.09), and at Tjärnö this was the case between midnight and sunset (ns). ALKBH9B exhibited a lower transcript level at solar noon and midnight, while peaking at sunset (Figure 5). Significant differences between latitudes were only detected at midnight (*p* < 0.05).

Interestingly, there was no effect of latitude on the global N^6^-methyl-adenosine (m^6^A) level, as exactly the same daily pattern was found in both populations of *Z. marina*. Although the effect of time was not significant (Table 5), there was a tendency for the m^6^A level to increase after sunset, peaking at midnight (Figure 5). This would nicely match with the gene-expression patterns that peaked at sunset, thus anticipating the m^6^A-methylation peaks and highlighting the time frame required for the translation and protein synthesis of the specific enzymes (Figure 5).

The similar m^6^A pattern and MTA/B daily expression levels observed here across *Z. marina* populations exposed to such different photoperiods was unexpected. Differences in the daily gene-expression regulation of definite gene groups (e.g., those associated with the photosynthetic process) were indeed detected at much smaller scales in seagrasses, i.e., in single *P. oceanica* populations extending along a depth gradient [57]. However, results from *A. thaliana* collected at diverse geographic locations revealed that the m^6^A level was not in direct relation with local environmental factors (e.g., PAR values), as the methylation level in mRNAs was relatively stable across accessions and the modification sites were quite conserved [21]. Nevertheless, the modification fraction at each site could vary depending on environmental factors [21]. Thus, our findings could indicate that the circadian timing of m^6^A oscillations could be largely driven by endogenous species-specific factors more than by external environmental factors. It would be interesting to address this topic in future studies and also to focus on other environmental gradients, such as, for instance, the bathymetric gradient, since variations in light quality and quantity along the water column are of fundamental importance for marine plants’ physiology and metabolism [62].

## 3. Materials and Methods

### 3.1. Inventory of RNA-Methylation-Associated Genes (m^6^A) in Seagrasses and Orthologs/Paralogs Dataset Construction

Protein collections for *C. nodosa*, *A. thaliana*, *O. sativa,* and *Z. marina* were downloaded from ENA [58], TAIR [63], TIGR [64], and ORCAE [50] databases, respectively. The networks of orthologs and paralogs were predicted by using a python software developed according to the pipeline described in two works that were already published [65,66]. Orthologs and paralogs, inferred by all-versus-all BLAST similarity searches [67] together with a Bidirectional Best Hit approach, were grouped into networks of proteins by using the NetworksX package [68]. A filtering step was introduced to select only the E-value cutoff that maximised the number of paralog networks for each species [69]. The graphical visualisation of the networks was obtained by Cytoscape software [70].

The function annotation was confirmed for each protein collection by performing a BLASTp analysis [67] against the SwissProt database (version of September 2019 [71]) and retrieving the best hit for each match. Proteins associated with an RNA-methylation function (i.e., m^6^A “writers” and “erasers”) were selected together with the related networks. The redundancy of *C. nodosa* sequences belonging to the selected networks was checked by the CAP3 software [72] and Clustal Omega [73] alignments software.

Protein sequences of the selected networks were aligned with MAFFT v7.397 (default parameters) [74], and alignments were cleaned with TrimAl v1.4 [75] using a 0.25 gap threshold, 0.25 residue overlap threshold, and 60% sequence overlap. The best-fit model of molecular evolution for each dataset was selected with ModelFinder [76] and implemented in IQ-TREE v. 1.6.11 [77] using the corrected Akaike Information Criterion (AICc). Maximum-likelihood trees were constructed with FastTree v2.1.11 [78] using four rounds of minimum-evolution nearest neighbour interchanges (NNIs). The graphical visualisation of the generated trees was obtained with the Interactive tree of life (iTOL) online tool [79].

### 3.2. Study Site, Plant Sampling, and Experimental Design

In May 2019 (8–10), specimens of *Cymodocea nodosa* and *Zostera marina* were collected from a shallow-water (1–2-m depth) mixed meadow near Ilha da Culatra in the Ria Formosa lagoon (Faro, Portugal, 37°1′9.6816′′ N, 7°55′49.584′′ W). Special care was taken to limit the breakage of rhizome connections. Plants were quickly transported to the Ramalhete field station (CCMAR, Centre of Marine Sciences) in darkened containers filled with seawater. Within a few hours after uprooting, the plants were transplanted to six 150-L cylindrical tanks located in an outdoor mesocosm facility. The bottom of each tank was covered with ca. 7 cm washed beach sand for adequate root and rhizome anchorage. Tanks were supplied with running seawater from Ria Formosa, previously filtered with sand and UV filters, in an independent open circuit configuration. Seagrass shoots were randomly assigned to each tank, in which they were then carefully planted (ca. 21 shoots per tank for *Z. marina*, and ca. 25 shoots per tank for *C. nodosa*). As the CCMAR field station and the collection site were geographically very close, the daily irradiance variations experienced by the plants in the field were identical to those experienced in the tanks. Before sampling, plants were left to recover from transportation and were monitored for 24 h after transplantation (i.e., tank acclimation). At the end of the acclimation period, leaf sections of *C. nodosa* and *Z. marina* were collected six times during a 24-h cycle, i.e., at midnight (0:00), dawn (04:30), sunrise (06:30), solar noon (13:00), sunset (20:30), and dusk (22:00).

Specimens of *Z. marina* were also collected from a shallow-water meadow (1–2-m depth) near the Tjärnö Marine Laboratory (Strömstad, Sweden, 58°52′25.752′′ N, 11°9′9.216′′ E) in mid-June 2019 (16–18). Plants were collected by snorkelling four times during a 24-h cycle, i.e., at midnight (01:15), sunrise (04:00), solar noon (13:15), and sunset (22:30). At this latitude, dusk and dawn do not occur in June due to the summer photoperiodic conditions. All information about sunrise/sunset times for the different latitudes were taken from https://www.timeanddate.com/.

Three biological replicates were collected for each species and time point (*n* = 3) at both localities. All leaf samples were rapidly cleaned of epiphytes and entirely submerged in an RNAlater© tissue collection (Ambion, Life Technologies, Waltham, MA USA) to inhibit RNA degradation. After allowing the solution to penetrate through the tissue for one night at 4 °C, the leaf samples were stored at −20 °C until RNA extraction.

### 3.3. RNA Extraction and cDNA Synthesis

The total RNA from the youngest fully developed leaves of *C. nodosa* and *Z. marina* was extracted with the Aurum™ Total RNA Mini Kit (BIO-RAD, Hercules, CA, USA.), following the manufacturer’s protocol. About 5–7 cm-long leaf sections were ground to a fine powder with a mortar and pestle containing liquid N2. 700 μL of lysis solution (supplemented with 2% (*w*/*v*) polyvinylpyrrolidone-40 (PVP) and 1% β-mercaptoethanol) were added to about 70–100 mg of powdered tissue. Samples were homogenized through a Mixer Mill MM300 (QIAGEN, Hilden, Germany) and tungsten carbide beads (3 mm) for 3 min at 20.1 Hz. The quality and purity (absence of DNA and protein contaminations) of the total RNA was checked using NanoDrop (ND-1000 UV-Vis spectrophotometer; NanoDrop Technologies, Wilmington, DE, USA) and 1% agarose gel electrophoresis. RNA was used when Abs260 nm/Abs280 nm and Abs260 nm/Abs230 nm ratios were >1.8 and 1.8 < x < 2, respectively. The RNA concentration was accurately determined by the Qubit^TM^ RNA BR Assay kit (Thermo Fisher Scientific, Waltham, MA USA) using the Qubit 2.0 Fluorometer (Thermo Fisher Scientific).

### 3.4. Primer Design, cDNA Synthesis, and Reverse-Transcription Quantitative Polymerase Chain Reaction (RT-qPCR)

The total RNA (500 ng) from each sample of *C. nodosa* and *Z. marina* was retro-transcribed into cDNA with the iScript™ cDNA synthesis kit (BIO-RAD), according to the manufacturer’s protocol. Primers for core components of the writer complex (MTA and MTB) and one eraser protein (ALKBH9B) were designed based on orthologous sequences found in the species of interest (see above) with the primer analysis software Primer3 v. 0.4.0 [80,81]. The design conditions included the primer length (18–23 bp), Tm (~60 °C), GC content (≥50%), and product size (100–250 bp). To normalise the target gene expression, we used the eukaryotic initiation factor 4A (eIF4A) as a reference gene, which was previously demonstrated to exhibit a stable expression along a daily cycle in *Z. marina* [82] and in other studies under different conditions and species, including *C. nodosa* [58,83,84,85,86]. The RT-qPCR efficiencies for all primer pairs were calculated from the slopes of the standard curves (with at least five dilution points) of the threshold cycle (CT) vs. cDNA concentration with the equation E = 10^−1/slope^.

RT-qPCR reactions were carried out in MicroAmp Optical 384-well reaction plates (Applied Biosystems, Foster City, CA, USA.) and the Viia7 Real Time PCR System (Applied Biosystems) and consisted of 5 μL of Fast SYBR^®^ Green Master Mix (Applied Biosystems), 1 μL of cDNA (1:10 and 1:5 diluted for *C. nodosa* and *Z. marina*, respectively) template, and 4 μL of 0.7 pmol μL^−1^ primers. The thermal profile of the reactions was as follows: 95 °C for 20 s, 40 times 95 °C for 1 s, and 60 °C for 20 s. To determine the specificity of the reaction, the melting curve of each amplicon from 60 to 95 °C was also detected. All RT-qPCR reactions were conducted in triplicate, and each assay included three no-template negative controls. The relative quantification of the transcript levels was obtained following [87]. In detail, the negative differences in the cycles to cross the threshold value between the RG and the respective GOI (−ΔCT) were calculated according to Equation (1). Mean −ΔCT values were then calculated for biological replicates at each time point (*n* = 3) from individual −ΔCT values:−ΔCT = CT_RG_ − CT_GOI_(1)

### 3.5. Global N^6^-Methyl-Adenosine (m^6^A) Quantification

The global RNA methylation (m^6^A) level was determined by an ELISA-like reaction with the EpiQuik™ m^6^A RNA Methylation Quantification Kit (Colorimetric) (Epigentek Inc., Farmingdale, NY USA), following the manufacturer’s instructions. Total RNA (100 ng) from three biological replicates and two technical replicates per species were employed for the analysis. To accurately determine the absolute amount of m^6^A in our samples, a standard curve was built in duplicates with at least four dilution points of a positive control and a negative control, both provided by the kit. The absorbance (OD) at 450 nm was assayed using a Multiskan™ FC Microplate Photometer (Thermo Fisher Scientific). The percentage (%) of m^6^A in our RNA samples was obtained using the following Formulas (2) and (3):m^6^A (ng) = (sample OD − NC OD)/(slope of the standard curve)(2)
m^6^A% = (m^6^A ng/ng input RNA sample) × 100%(3)

### 3.6. Statistical Analysis

A multivariate analysis was first used to assess the overall signal of all GOIs (using −ΔCT values). Specifically, a two-way permutational multivariate analysis of variance (PERMANOVA) was conducted with the Primer 6 v.6.1.12 & PERMANOVA + v.1.0.2 software package (PRIMER-E Ltd. Albany, Auckland, New Zealand) [88] to test for intra- and interspecific differences in the transcript levels of GOIs along the diel cycle and between different latitudes. The “interspecific” analysis consisted of two fixed factors: Species (Sp) with two levels (i.e., *C. nodosa* and *Z. marina*) and Time (Ti) with six levels (i.e., midnight, dawn, sunrise, solar noon, sunset, and dusk). The “intraspecific” analysis consisted of two fixed factors: Latitude (Lat) with two levels (i.e., Faro—Portugal and Tjärnö—Sweden) and Time (Ti) with four levels (i.e., midnight, sunrise, solar noon, and sunset). Subsequently, a univariate analysis (two-way ANOVA), was used to assess the contribution of individual GOIs and the global m^6^A level to the observed effects. Two-way ANOVAs were performed using the statistical package STATISTICA (StatSoft, Inc. v. 10, Brookline, MA, USA) and consisted of the same factors and levels as outlined above. Data normality was tested using the Shapiro–Wilk test, and the variance homogeneity was verified using Levene’s test. The Student–Newman–Keuls (SNK) post-hoc test was used whenever significant differences were detected. Pearson’s correlation analyses were also conducted across gene-expression patterns using the statistical package STATISTICA (StatSoft, Inc. v. 10).

## 4. Conclusions

Here, we identified orthologs and paralogs of RNA-methylation related genes (i.e., m^6^A writers and erasers) through a comparative approach that involved two important marine plants (*Z. marina* and *C. nodosa*), together with the model terrestrial species *O. sativa* and *A. thaliana*. Seagrasses possess the entire repertoire of m^6^A writers and eraser genes (i.e., MTA/B, FIP37, HAKAI, VIRILIZER, and ALKBH9B/10B); however, an expansion of the FIP37 gene family was observed at the gene (in *Z. marina*) and transcript levels (in *C. nodosa*). The transcriptome of *C. nodosa* was particularly redundant in transcripts encoding for eraser proteins of the Alk family. These sequences probably represent different isoforms and/or transcript variants. The reason for such a redundancy and the functional role of these variants is currently unknown, but it is worth investigating in the future.

Our inter- and intraspecific analyses of the expression patterns of a subgroup of writers and erasers along a 24-h cycle revealed a striking rhythmicity of such transcripts and concomitant m^6^A levels in *C. nodosa* and *Z. marina*. The “circadian effect” could be bidirectional, as these daily oscillations of mRNA levels suggest that those genes could actually influence circadian components or could be themselves regulated by the circadian-clock system. Interestingly, in both species, m^6^A methylation and the abundance of related transcripts always peaked toward the dark period, from sunset to midnight. This might suggest a role of mRNA m^6^A modifications in driving specific transcriptome rearrangements [21] during the night phase. In fact, as demonstrated in some terrestrial species, a massive gene expression induction often occurs towards and during the dark phase, involving specific gene sets [89]. Although the daily patterns were asynchronous between species at the same geographic location, they were very similar in *Z. marina* populations at contrasting latitudes. These observations suggest that the m^6^A RNA-methylation rhythms observed here are regulated by endogenous species-specific cues more than by external environmental stimuli [90] such as light (PAR) intensities.

Our results suggest that RNA methylation might widely participate in the daily regulation of gene expression and circadian-clock functions in seagrasses, potentially affecting the photo-biological behaviour of the species and their ability to spread across different latitudes and photoperiodic regimes. Further analysis could provide valuable information on the composition, function, and evolution of m^6^A gene families in marine plants. Moreover, future studies should be carried out to obtain the transcriptome-wide map of m^6^A marks in seagrasses, thus identifying specific genes whose expression can be altered by such modifications. Exploring the presence of m^6^A marks on circadian clock-related genes, or on those involved in fundamental metabolic routes such as photosynthesis and chloroplast functions, would be a great step forward into recognising the roles of m^6^A modifications and their relevance to plant physiological rhythms and environmental adaptation.

## Figures and Tables

**Figure 1 ijms-21-07508-f001:**
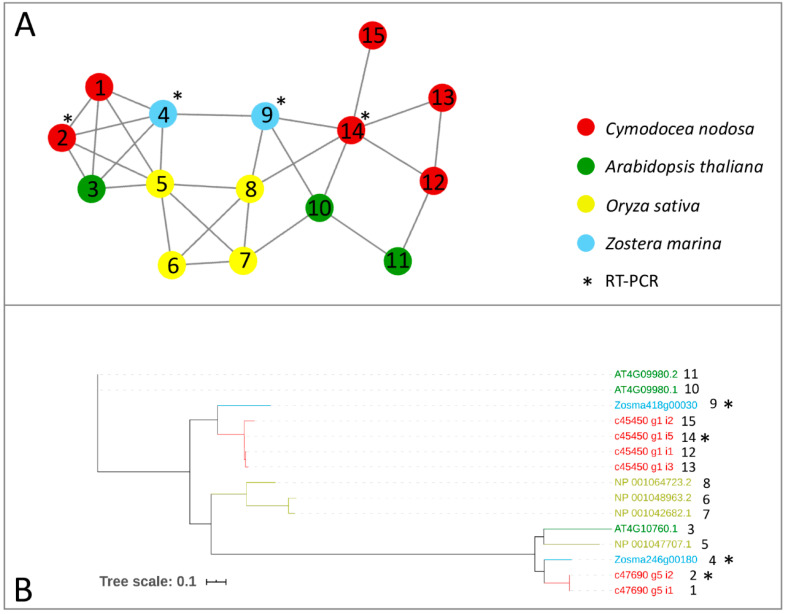
Orthologs/paralogs networks of writers: (**A**) MTA and MTB writers (NET_3623). Circles represent proteins per species, according to the legend. Grey lines represent homology relationships. Transcript sequences analysed by RT-qPCR are marked with an asterisk; (**B**) Phylogenetic tree of MTA and MTB writers, constructed using an Approximately Maximum-Likelihood algorithm.

**Figure 2 ijms-21-07508-f002:**
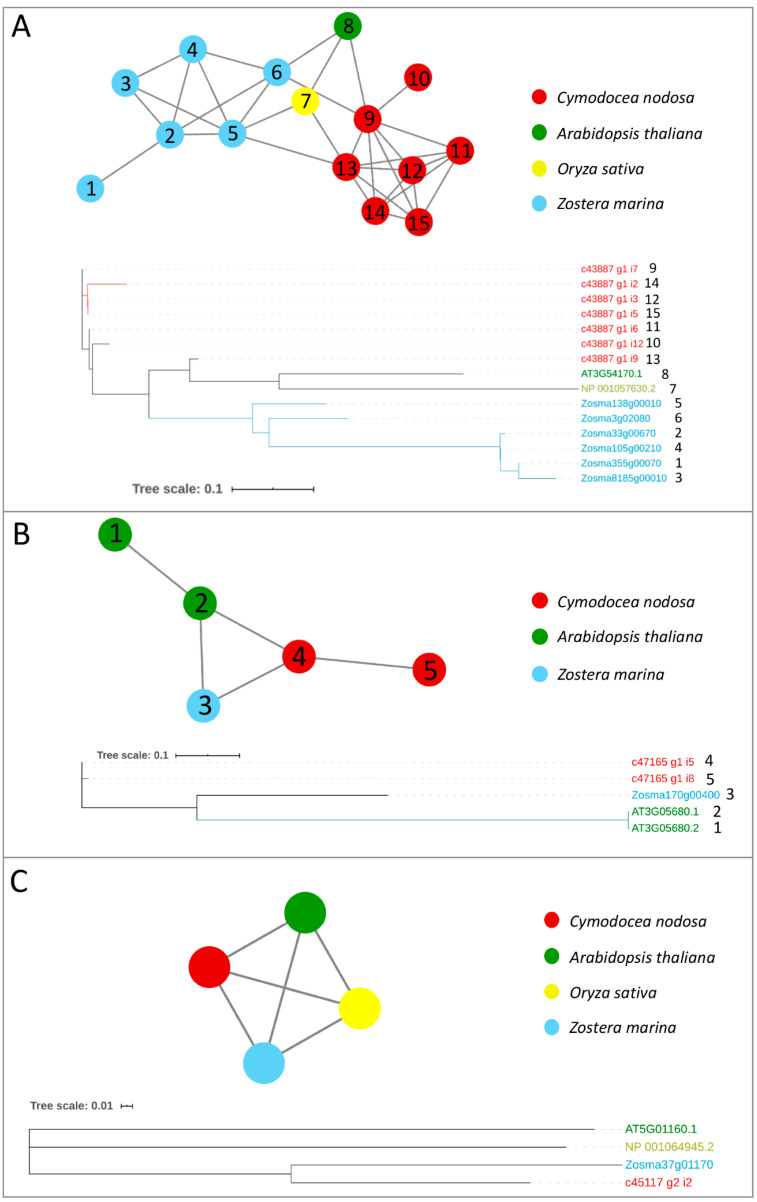
Orthologs/paralogs networks and phylogenetic trees of additional writers: (**A**) FIP37 (NET_2064); (**B**) VIRILIZER (NET_1791); and (**C**) HAKAI (NET_8223). Circles represent proteins per species, according to the legend. Grey lines represent homology relationships. Phylogenetic trees are constructed using an Approximately Maximum-Likelihood algorithm

**Figure 3 ijms-21-07508-f003:**
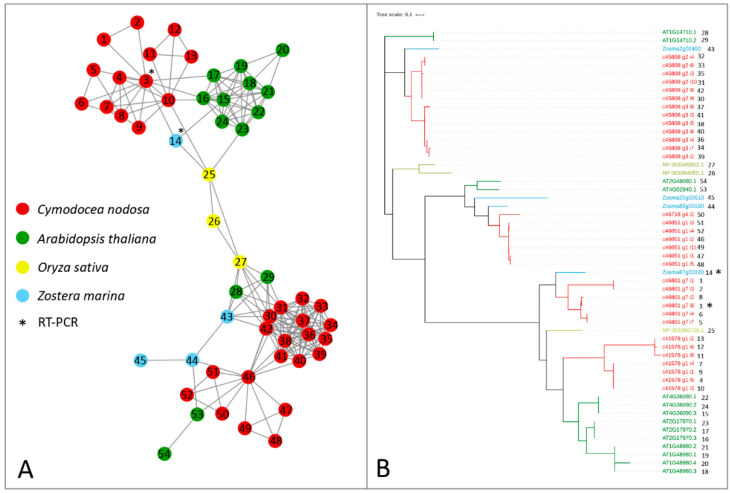
Orthologs/paralogs networks of erasers: (**A**) ALKBH9B and ALKBH10B (NET_160). Circles represent proteins per species, according to the legend. Grey lines represent homology relationships. Transcript sequences analysed by RT-qPCR are marked with an asterisk; (**B**) Phylogenetic tree of ALKBH9B and ALKBH10B erasers, constructed using an Approximately Maximum-Likelihood algorithm.

**Figure 4 ijms-21-07508-f004:**
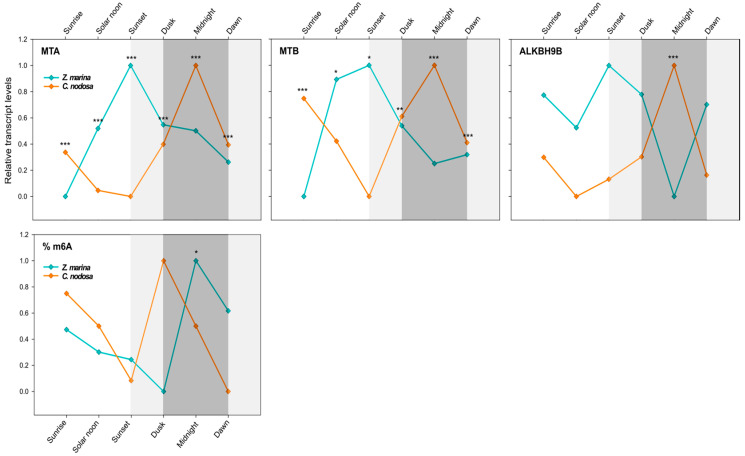
Daily transcript levels (as normalised −ΔCT data) of writer (MTA and MTB) and eraser (ALKBH9B) genes and % of global m^6^A (normalised data) in *Z. marina* and *C. nodosa* (interspecific analysis) at Ria Formosa lagoon (Faro, Portugal). Asterisks represent significant differences between species at the specific time points, as resulting from SNK post-hoc tests following a two-way ANOVA. Light and dark grey boxes highlight the twilight and night times, respectively. * *p* < 0.05; ** *p* < 0.01; *** *p* < 0.001.

**Figure 5 ijms-21-07508-f005:**
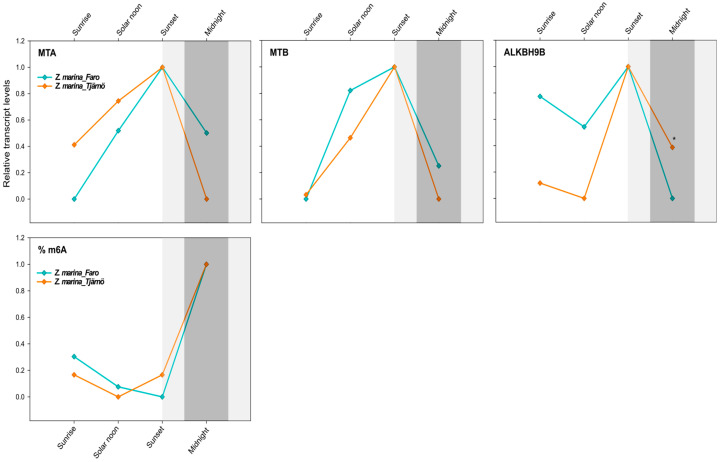
Daily transcript levels (as normalised −ΔCT data) of writer (MTA and MTB) and eraser (ALKBH9B) genes and % of global m^6^A (normalised data) in *Z. marina* at Ria Formosa lagoon, Faro—Portugal (Lat. 37.019356) and at Tjärnö, Strömstad—Sweden (Lat. 58.879724) (intraspecific analysis). Asterisks represent significant differences between species at the specific time points, as resulting from SNK post-hoc tests following a two-way ANOVA. Light and dark grey boxes highlight the twilight and night times, respectively. * *p* < 0.05.

**Table 1 ijms-21-07508-t001:** List of writers and erasers involved in RNA methylation (m^6^A) in terrestrial model plants (*A. thaliana* and *O. sativa*) and their orthologs in the seagrass *Z. marina* and *C. nodosa*, as resulting from the gene-network analysis. The ID refers to transcripts in *Arabidopsis*, *Zostera* and *Cymodocea*, or to proteins in *Oryza*.

Type	Gene Name	*Arabidopsis thaliana*		*Oryza sativa*		*Zostera marina*		*Cymodocea nodosa*		
		N° Proteins	ID	N° Proteins	ID	N° Proteins	ID	N° Proteins	ID	Network
Writers	MTA	1	AT4G10760.1	1	NP_001047707.1	1	Zosma246g00180	2	c47690_g5_i2	NET_3623
									c47690_g5_i1	NET_3623
Writers	MTB	2	AT4G09980.2	3	NP_001064723.2	1	Zosma418g00030	4	c45450_g1_i3	NET_3623
			AT4G09980.1		NP_001048963.2				c45450_g1_i5	NET_3623
					NP_001042682.1				c45450_g1_i1	NET_3623
									c45450_g1_i2	NET_3623
Writers	FIP37	1	AT3G54170.1	1	NP_001057630.2	6	Zosma8185g00010	7	c43887_g1_i12	NET_2064
							Zosma355g00070		c43887_g1_i2	NET_2064
							Zosma33g00670		c43887_g1_i6	NET_2064
							Zosma3g02080		c43887_g1_i7	NET_2064
							Zosma138g00010		c43887_g1_i5	NET_2064
							Zosma105g00210		c43887_g1_i9	NET_2064
									c43887_g1_i3	NET_2064
Writers	VIRILIZER	2	AT3G05680.1	-	-	1	Zosma170g00400	3	c47165_g1_i3	NET_1791
			AT3G05680.2						c47165_g1_i8	NET_1791
									c47165_g1_i5	NET_1791
Writers	HAKAI	1	AT5G01160.1	1	NP_001064945.2	1	Zosma37g01170	1	c45117_g2_i2	NET_8223
Erasers	ALKBH9B	10	AT4G36090.2	1	NP_001056738.1	1	Zosma87g00220	13	c41578_g1_i1	NET_160
			AT1G48980.1						c41578_g1_i2	NET_160
			AT4G36090.3						c41578_g1_i3	NET_160
			AT1G48980.4						c41578_g1_i4	NET_160
			AT1G48980.2						c41578_g1_i5	NET_160
			AT2G17970.2						c41578_g1_i6	NET_160
			AT2G17970.3						c41578_g1_i8	NET_160
			AT2G17970.1						c46801_g7_i1	NET_160
			AT1G48980.3						c46801_g7_i2	NET_160
			AT4G36090.1						c46801_g7_i3	NET_160
									c46801_g7_i4	NET_160
									c46801_g7_i7	NET_160
									c46801_g7_i8	NET_160
Erasers	ALKBH10B	4	AT1G14710.2	2	NP_001064055.1	3	Zosma89g00160	20	c45808_g2_i10	NET_160
			AT1G14710.1		NP_001049502.1		Zosma25g00610		c45808_g2_i3	NET_160
			AT4G02940.1				Zosma2g02460		c45808_g2_i4	NET_160
			AT2G48080.1						c45808_g2_i6	NET_160
									c45808_g2_i8	NET_160
									c45808_g2_i9	NET_160
									c45808_g3_i2	NET_160
									c45808_g3_i3	NET_160
									c45808_g3_i5	NET_160
									c45808_g3_i6	NET_160
									c45808_g3_i7	NET_160
									c45808_g3_i8	NET_160
									c45808_g3_i4	NET_160
									c46051_g1_i1	NET_160
									c46051_g1_i11	NET_160
									c46051_g1_i2	NET_160
									c46051_g1_i3	NET_160
									c46051_g1_i4	NET_160
									c46051_g1_i5	NET_160
									c46718_g4_i2	NET_160

**Table 2 ijms-21-07508-t002:** List of Genes of Interest (GOIs) in *C. nodosa* and *Z. marina* assessed by RT-qPCR. The gene acronym, protein name, species, primer sequences, amplicon size (S, bp), percent efficiency (*E*), correlation coefficient (R^2^), ID, and network are given.

Gene Acronym	Protein	Species	Primer Sequences 5′→3′	S	*E*	R^2^	ID	Network
MTA	N^6^-adenosine-methyltransferase MT-A70-like	*C. nodosa*	F: GGGGCAGTTTGGGGTTATTA R: GCTCGTCCAGTTACCCAAAG	150	100%	0.99	c47690_g5_i2	NET_3623
MTB	N^6^-adenosine-methyltransferase non-catalytic subunit MTB	*C. nodosa*	F: CCTTGGGAGGAGTATGTCCA R: GCAAACTTGGAGTGGCATTT	244	100%	0.99	c45450_g1_i5	NET_3623
MTA	N^6^-adenosine-methyltransferase MT-A70-like	*Z. marina*	F: TTATGGCAGATCCACCTTGG R: GCTCGTCCAGTTACCCAAAG	132	100%	0.99	Zosma246g00180	NET_3623
MTB	N^6^-adenosine-methyltransferase non-catalytic subunit MTB	*Z. marina*	F: CTCCATAGAGCTCCTGGTTCTG R: ACACTGCCTACCCTGCTCAA	150	100%	0.98	Zosma418g00030	NET_3623
ALKBH9B	RNA demethylase ALKBH9B	*C. nodosa*	F: ATCGGTCAGTTGGGATGAAG R: AACTCGTACACACAATTCAC	225	100%	0.99	c46801_g7_i8	NET_160
ALKBH9B	RNA demethylase ALKBH9B	*Z. marina*	F: ACGACTTTGTCCGACCCTTC R: GAACACCTGGGATGCAATGC	189	90%	0.99	Zosma87g00220	NET_160

**Table 3 ijms-21-07508-t003:** Results of two-way PERMANOVAs conducted on multivariate gene-expression data (−ΔCT values for all GOIs) for inter- (*C. nodosa* vs. *Z. marina*) and intraspecific (*Z. marina* at different latitudes) comparisons. *p*_(perm)_ < 0.05 are in bold; *p*_(MC)_ < 0.01 are indicated in brackets.

Table PERMANOVAs							
				Pseudo-		Unique	
Source	df	SS	MS	F	*P* _(perm)_	perms	Pairwise Tests (*P*_MC_)
*Inter-specific analysis*							
Species (Sp)	1	252.64	252.64	172.14	**0.0001**	9919	*Z. marina*: Sunrise ≠ Sunset; Sunrise ≠ dusk; Dawn = Sunset (*p* = 0.06); Sunrise = Solar noon (*p* = 0.08) *C. nodosa*: Sunrise ≠ Solar noon; Sunrise ≠ Sunset; Midnight = Solar noon (*p* = 0.06); Midnight = Sunset (*p* = 0.06)
Time (Ti)	5	11.525	2.305	1.5705	0.1633	9942	
Sp × Ti	5	28.621	5.7242	3.9002	**0.0038**	9941	
Res	24	35.224	1.4677				
Total	35	328.01					
*Intra-specific analysis*							
Latitude (La)	1	7.0596	7.0596	3.9333	**0.0328**	9963	*Z. marina* Faro: Sunrise ≠ Sunset; Sunrise = Solar noon (*p* = 0.08) *Z. marina* Tjärnö: Midnight = Sunset (*p* = 0.08)
Time (Ti)	3	22.844	7.6147	4.2426	**0.0124**	9956	
La × Ti	3	6.1978	2.0659	1.1511	0.3495	9946	
Res	16	28.717	1.7948				
Total	23	64.819					


df: degree of freedom, SS: sum of squares, MS: mean squares, Pseudo-F: pseudo-F ratio.

**Table 4 ijms-21-07508-t004:** Results of two-way ANOVAs conducted on −ΔCT values of individual GOIs for inter- (*C. nodosa* vs. *Z. marina*) and intraspecific (*Z. marina* at different latitudes) comparisons. *p* < 0.05 are in bold; *p* < 0.1 are underlined.

Two-Way ANOVAs									
Effect	*df*	Interspecific Analysis			Effect	*df*	Intraspecific Analysis		
		MS	F	*p*			MS	F	*p*
MTA					MTA				
Species (Sp)	1	173.420	289.482	**0.000**	Latitude (La)	1	0.086	0.111	0.743
Time (Ti)	5	1.373	2.292	0.077	Time (Ti)	3	3.737	4.841	**0.014**
Sp × Ti	5	2.557	4.269	**0.006**	La × Ti	3	1.111	1.439	0.269
Error	24	0.599			Error	16	0.772		
MTB					MTB				
Species (Sp)	1	73.936	135.660	**0.000**	Latitude (La)	1	0.584	0.812	0.381
Time (Ti)	5	0.593	1.089	0.392	Time (Ti)	3	3.077	4.274	**0.021**
Sp × Ti	5	1.534	2.814	**0.039**	La × Ti	3	0.594	0.825	0.499
Error	24	0.545			Error	16	0.720		
ALKBH9B					ALKBH9B				
Species (Sp)	1	5.2886	16.343	**0.000**	Latitude (La)	1	6.389	21.081	**0.000**
Time (Ti)	5	0.3386	1.046	0.414	Time (Ti)	3	0.801	2.642	0.085
Sp × Ti	5	1.6332	5.047	**0.003**	La × Ti	3	0.361	1.192	0.344
Error	24	0.3236			Error	16	0.303		

df: degree of freedom, MS: mean squares, F: F ratio.

**Table 5 ijms-21-07508-t005:** Results of two-way ANOVAs conducted on the global N^6^-methyl-adenosine (m^6^A) level for inter- (*C. nodosa* vs. *Z. marina*) and intraspecific (*Z. marina* at different latitudes) comparisons. *p* < 0.05 are in bold.

Two-Way ANOVAs									
Effect	*df*	Interspecific Analysis			Effect	*df*	Intraspecific Analysis		
		MS	F	*p*			MS	F	*p*
m^6^A					m^6^A				
Species (Sp)	1	0.001	17.763	**0.000**	Latitude (La)	1	0.305	2.297	0.149
Time (Ti)	5	0.000	1.274	0.309	Time (Ti)	3	0.124	0.936	0.446
Sp × Ti	5	0.000	1.851	0.143	La × Ti	3	0.068	0.511	0.680
Error	23	0.000			Error	16	0.133		

df: degree of freedom, MS: mean squares, F: F ratio.

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
