# Peer review of "m6A RNA Methylation in Marine Plants: First Insights and Relevance for Biological Rhythms"

_ijms, 2020, doi:10.3390/ijms21207508_

Round 1
Reviewer 1 Report
The authors address a topic of epitranscriptomics in association with circadian rythms in two non-model marine plant species. They identify m6A-related genes in these seagrass species and analyse daily changes in transcript levels of two genes coding for writers and one coding for an eraser enzyme in both plants. They also examine changes in global levels of m6A in RNA in parallel. It is quite enigmatic that transcript levels of writers and erasers peak at the same time.
These results are novel in marine plants and may be of interest to readers which are otherwise left only with results from a few model plants.
After reading the MS, I was a bit disappointed that - contrary to my expectations raised by the title of the MS - no representative transcriptomic/epitranscriptomic data were generated or analysed in the MS. As the authors point out in their conclusion, future studies should be carried out to obtain the transcriptome-wide map of m6A marks in seagrasses, thus identifying specific genes whose expression can be altered by such modifications. While I can understand that such results are probably beyond the scope of the present study, at least a sample set of genes (maybe derived from data obtained in model plants) should be analysed for correlation between daily changes in their transcript levels, and m6A levels (e.g. using relative quantity of trancripts in total RNA and IP fraction of m6A-RNAs). Such result would greatly increase the impact of the manuscript. Of course, a transcriptome-wide comparison between such samples would be optimal. Otherwise, it is not possible to distinguish between a high level of m6A-modification of a few RNA species, or a global increase of m6A-RNAs.
Minor specific points:
- The authors should distinguish between the "expression levels" (which they do not actually analyse) and "transcript levels" (which are measured here) in the text and figures (y-axis description).
- Symbols in column headers listing statistical values in Tables 3-5 should be explained in the Table heading or a footnote.
Author Response
Review of manuscript ID: ijms-919817
We wish to thank the reviewer 1 for the comments on the manuscript “Epitranscriptomics in marine plants: first insights and relevance for biological rhythms”. The comments have been carefully considered and below we report a point-by-point response to the questions and/or suggestions.
Our notes to the comments are in italics.
REVIEWER 1
Comments and Suggestions for Authors
#1 The authors address a topic of epitranscriptomics in association with circadian rythms in two non-model marine plant species. They identify m6A-related genes in these seagrass species and analyse daily changes in transcript levels of two genes coding for writers and one coding for an eraser enzyme in both plants. They also examine changes in global levels of m6A in RNA in parallel. It is quite enigmatic that transcript levels of writers and erasers peak at the same time.
#1 These two classes of proteins act according to a complex interplay that does not preclude their simultaneous action. In addition, at the same time of the day, methylation and demethylation processes could involve different targets.
These results are novel in marine plants and may be of interest to readers which are otherwise left only with results from a few model plants.
#2 After reading the MS, I was a bit disappointed that - contrary to my expectations raised by the title of the MS - no representative transcriptomic/epitranscriptomic data were generated or analysed in the MS. As the authors point out in their conclusion, future studies should be carried out to obtain the transcriptome-wide map of m6A marks in seagrasses, thus identifying specific genes whose expression can be altered by such modifications. While I can understand that such results are probably beyond the scope of the present study, at least a sample set of genes (maybe derived from data obtained in model plants) should be analysed for correlation between daily changes in their transcript levels, and m6A levels (e.g. using relative quantity of trancripts in total RNA and IP fraction of m6A-RNAs). Such result would greatly increase the impact of the manuscript. Of course, a transcriptome-wide comparison between such samples would be optimal. Otherwise, it is not possible to distinguish between a high level of m6A-modification of a few RNA species, or a global increase of m6A-RNAs.
#2 We understand the doubts of the reviewer about the title, hence we changed from “Epitranscriptomics in Marine Plants: First Insights and Relevance for Biological Rhythms” into “m6A RNA methylation in Marine Plants: First Insights and Relevance for Biological Rhythms” in order to avoid readers thinking about the use of “omics” approaches in the manuscript.
Regarding the second point raised by the reviewer, we acknowledge the importance of the proposed experiment to increase the impact and the completeness of the paper. However, we believe such analysis is out of the scope of the current study. The specific goals of this paper were indeed 1) to identify for the first time gene families involved in m6A methylation in seagrasses, and reconstruct their phylogenetic relationships with terrestrial species; and 2) to understand if some of these genes exhibited a definite daily expression pattern, thus suggesting their involvement in the regulation of the circadian-clock system and vice versa (their regulation by circadian-clock components). We further wanted to assess the differences in the regulation of these genes across species and between latitudes, and this is another fundamental part of the manuscript. Currently, no information are available about targets of m6A methylation marks in seagrasses, while scarce information are available for terrestrial species. Therefore, we believe the suggested analysis, which would require a big effort, and overcomes the specific aims of this work, should to be performed in a separated work.
Minor specific points:
#1 The authors should distinguish between the "expression levels" (which they do not actually analyse) and "transcript levels" (which are measured here) in the text and figures (y-axis description).
#1 As suggested by the reviewer, we changed “expression levels” with “transcript levels” in both text and figures.
#2 Symbols in column headers listing statistical values in Tables 3-5 should be explained in the Table heading or a footnote.
#2 We explained ANOVA’s abbreviations in Table footnotes.
Reviewer 2 Report
This manuscript by Ruocco et al. describes the diurnal rhythm of RNA methylation in two seaweed species. It is a new and invaluable study that provides new insights into the field of circadian clock research.
Major comments.
None
Minor comments.
Although the introduction is comprehensive, it seems to be too much content for an introduction of an original paper. It would be better to make it more compact.
Some of the statements in the first three paragraphs of Results and Discussion (lines 136-152) overlap with the introduction. Even for those sentences that do not overlap, they do not include results and therefore should be included in the Introduction.
The authors should mention the recent paper by Fustin et al. (Commun Biol. 2020 May 6;3(1):211. doi: 10.1038/s42003-020-0942-0.) which shows the importance of RNA methylation in the circadian clock in several taxa.
Why does the expression of methylation-related genes not coincide with the actual phase of methylation? The authors should make more mention of this point in their discussion.
The conclusion section appears after the materials and methods section, but I am not sure if this is the journal's format.
Author Response
We wish to thank the reviewer 2 for the comments on our paper "Epitranscriptomics in marine plants: first insights and relevance for biological rhythms". We carefully addressed the comments/suggestions in the point by point response reported below.
Our notes to reviewer comments are in italics.
REVIEWER 2
Comments and Suggestions for Authors
This manuscript by Ruocco et al. describes the diurnal rhythm of RNA methylation in two seaweed species. It is a new and invaluable study that provides new insights into the field of circadian clock research.
Major comments.
None
Minor comments.
#1 although the introduction is comprehensive, it seems to be too much content for an introduction of an original paper. It would be better to make it more compact.
#1 We appreciate the reviewer suggestion to shorten the introduction section and make it more compact. See e.g. LINES 74-75, 79-83 and 90-91.
#2 Some of the statements in the first three paragraphs of Results and Discussion (lines 136-152) overlap with the introduction. Even for those sentences that do not overlap, they do not include results and therefore should be included in the Introduction.
#2 As suggested by the reviewer, we removed the sentences in question (LINES 136-152) as such concepts are already included in the introduction section.
#3 The authors should mention the recent paper by Fustin et al. (Commun Biol. 2020 May 6;3(1):211. doi: 10.1038/s42003-020-0942-0.) which shows the importance of RNA methylation in the circadian clock in several taxa.
#3 We thank the reviewer for mentioning this very recent paper by Fustin et al. We included it in our reference list (LINE 96).
#4 Why does the expression of methylation-related genes not coincide with the actual phase of methylation? The authors should make more mention of this point in their discussion.
#4 In general, a match between maximum transcript expression of m6a-related genes and actual increase in RNA m6a methylation should not be expected. The increase in transcript abundance, translation and synthesis of the enzymes that will carry out the enzymatic reaction (e.g. adding methyl groups on RNAs) require a certain time. This is well visible e.g. in Z. marina, where max. transcript abundance of MTA and MTB occurs at sunset, while max % methylation occurs at midnight. However, this pattern is not always respected, as max. m6A also results from the activity of demethylases enzymes, and other factors as the half-life of enzymes etc. We added few lines on this in LINES 346-347.
#5 The conclusion section appears after the materials and methods section, but I am not sure if this is the journal's format.
#5 We just followed the journal format and the conclusion section is included after material and methods.
Round 2
Reviewer 1 Report
The authors have adapted the title of the manuscript to be more relevant to its contents. They also corrected minor errors and shortcomings. Although they did not perform the suggested additional experiment which could potentially increase the impact of the study, I believe the MS is now acceptable for publication.